

# Security awareness of single sign-on account in the academic community: the roles of demographics, privacy concerns, and Big-Five personality

Ahmad R. Pratama[1], Firman M. Firmansyah[2] and Fayruz Rahma[1]

[1] Department of Informatics, Universitas Islam Indonesia, Sleman, Daerah Istimewa Yogyakarta, Indonesia
[2] Department of Technology and Society, Stony Brook University, Stony Brook, NY, United States of America

## ABSTRACT

Single sign-on (SSO) enables users to authenticate across multiple related but independent systems using a single username and password. While the number of higher education institutions adopting SSO continues to grow, little is known about the academic community's security awareness regarding SSO. This paper aims to examine the security awareness of SSO across various demographic groups within a single higher education institution based on their age, gender, and academic roles. Additionally, we investigate some psychological factors (i.e., privacy concerns and personality traits) that may influence users' level of SSO security awareness. Using survey data collected from 283 participants (faculty, staff, and students) and analyzed using a hierarchical linear regression model, we discovered a generational gap, but no gender gap, in security awareness of SSO. Additionally, our findings confirm that students have a significantly lower level of security awareness than faculty and staff. Finally, we discovered that privacy concerns have no effect on SSO security awareness on their own. Rather, they interact with the user's personality traits, most notably agreeableness and conscientiousness. The findings of this study lay the groundwork for future research and interventions aimed at increasing cybersecurity awareness among users of various demographic groups as well as closing any existing gaps between them.

## INTRODUCTION

Single sign-on (SSO) is a cybersecurity measure that enables the use of a single username and password to authenticate the same user across multiple related but independent networks, computers, or information systems without having users enter their authentication credentials multiple times. SSO enables end users to increase their productivity by significantly reducing the time required for authentication processes (*James et al., 2020*). On the other hand, it may also result in financial savings for the institution that implements it, for example, through cost savings in their information technology expenditures (*Chinitz, 2000*; *Gellert et al., 2019*; *D'costa-Alphonso & Lane, 2010*). However, it was not until the late 2000s that SSO adoption became more widespread in a wide variety of organizations and

Corresponding author
Ahmad R. Pratama,
ahmad.rafie@uii.ac.id

enterprises, mainly due to performance issues (*D'costa-Alphonso & Lane, 2010*) or insecure implementation (*Bai et al., 2013*) in its early days. Nonetheless, SSO adoption was not uniform across sectors and regions of the world. Even in the early 2010s, some people still refused to adopt SSO because they did not perceive an urgent need for it although that perception changed as SSO's design and implementation improved (*Sun et al., 2011*).

There are numerous types of information systems in the academic community, particularly in higher education institutions, including learning management systems (LMS), academic information systems (AIS), management information systems (MIS), and payroll services. In general, there are at least three distinct academic roles (*i.e.,* students, faculty, and staff), each of which usually has a different type and level of access to the institution's information systems. Historically, colleges and universities required users to have separate accounts for each system. This situation resulted in significant frustration for the users and increased support costs for the institution. Implementing SSO resolves that issue by allowing users to log in to all systems using the same username and password.

Along with the conveniences that SSO provides, there is an arguably greater risk associated with the fact that that same account now has access to everything the user has access to. If attackers gain access to an SSO account, they have the potential to cause additional damage, not just to the user whose SSO account was compromised, but also to other users and the institution itself. Even more so now that many universities have been forced to fully embrace online education in the aftermath of the COVID-19 pandemic, forcing everyone, including those with limited online experience, to quickly adapt to this digital transformation, the risk has increased. As a result, safeguarding SSO accounts is becoming increasingly critical, even if not all users are aware of such issues.

Numerous studies have been conducted on security awareness, including in the academic community. However, little is known about the use of SSO in the academic community in general, and even more specifically about users' security awareness regarding their SSO accounts. This study aims to determine the level of security awareness among members of the academic community regarding SSO accounts. We are particularly interested in examining the psychological factors within individuals that can help predict their level of security awareness, specifically their privacy concerns and personalities, and determining whether there is any interaction between them. Additionally, we would like to determine whether the level of awareness varies by demographic characteristics (*e.g.*, age, gender) and academic roles (*i.e.,* student, faculty, and staff). To be more precise, the term "staff" here refers to administrative personnel, including librarians and technical support personnel, as opposed to non-administrative personnel such as janitorial and security personnel.

The following section discusses our theoretical framework, beginning with the relationship between security awareness, our dependent variable, and each of the independent variables, which include demographic variables, academic roles, familiarity with SSO, privacy concerns, and Big-Five personality traits. Following that, we discuss the research design in detail, including information about our participants, research instruments, and the data analysis procedures. Finally, we present the statistical analysis results before discussing the key findings and concluding by restating the main takeaways and identifying future research directions.

# THEORETICAL FRAMEWORK

## Demographics and security awareness

Among demographic variables, gender and age have been identified as significant factors that differentiate cyber security behaviors among users. For example, _Anwar et al. (2017)_ discovered that female users are more likely than male users to have behaviors that increase the likelihood of becoming a victim of cybercrimes. For example, they tend to reuse the same passwords across multiple social media accounts, open email attachments from unknown people, and click peculiar short URLs posted on the Internet. Meanwhile, _Grimes et al. (2010)_ discovered that older users are less familiar with cyber security measures (_e.g._, keeping their passwords private) and are less knowledgeable about cyber security risks (_e.g._, having difficulty in recognizing phishing, computer viruses, and spams). In another study, _Pratama & Firmansyah (2021)_ revealed that females and older users were less likely to be aware of, let alone adopt, two-factor authentication (2FA), making them particularly vulnerable to cyber security threats. Taking these findings into account, we hypothesize that:

H1: Females are less aware of SSO security
H2: Older people are less aware of SSO security

It is worth highlighting that, by no means, do we assume that being female and older in and of itself then makes people less aware of SSO security. Rather, in this study, we examine whether such associations, which does not necessarily mean causation, as shown in the literature between the respected demographic variables and security awareness. Still exist and if they are also true in the case of SSO security. Such significant findings will expose demographic gaps needing to be addressed by future research, for instance, on why the gaps keep occurring and how to close them.

## Academic roles and security awareness

Most studies in cybersecurity awareness and behaviors in the academic community tend to focus on either students (_Farooq et al., 2015_; _Ngoqo & Flowerday, 2015_; _Zwilling et al., 2022_) or faculty/staff (_Yerby & Floyd, 2018_) only. In one study involving both academic roles, faculty/staff reported higher security behaviors than students (_Gratian et al., 2018_). Taking that into account and due to the nature of the role that faculty and staff usually have more systems and data to access within an academic institution, and thus more to lose than students should their SSO accounts be compromised, we hypothesize that:

H3: Students are less aware of SSO security than faculty and staff

## SSO familiarity and security awareness

SSO adoption in higher education is relatively recent in comparison to other industrial and commercial organizations. In this particular institution where the study was conducted, the SSO system, managed and operated directly by the university's Board of Information Systems, was implemented using Shibboleth, an open-source SSO system based on Security Assertion Markup Language (SAML) protocol. The SSO was not fully implemented university-wide until 2019, just a few months before the onset of the COVID-19 pandemic. Taking that into account, we hypothesize that:

H4: Familiarity to SSO positively predicts SSO security awareness

## Privacy concerns and security awareness

Individual concerns over what, when, and how their private information is being shared to others when using information technology products and services have been widely discussed in the literature (*Petronio & Child, 2020*). For instance, multiple studies have revealed that the privacy paradox, discrepancy between stated privacy attitudes and actual privacy behaviors, in using the technologies does exist in various contexts and across cultures (*Aleisa, Renaud & Bongiovanni, 2020*; *Barth et al., 2019*; *Kokolakis, 2017*). Some argue that this phenomenon can be explained by privacy calculus, which reflects the discrepancy between anticipated risks and expected benefits associated with letting go of some private information (*Goad, Collins & Gal, 2021*). Should the benefits be higher, users tend to compromise their privacy as opposed to holding it in any other cases. These arguments suggest that privacy concerns lead to more cautious decisions in whether to use information technology related products or services, including SSO implementation as pointed out by some studies in the literature (*Cho, Kim & Sundar, 2020*; *Heckle & Lutters, 2007*). Bringing all those findings to the current study's context, we predict that:

H5: Privacy concerns positively predict SSO security awareness

## Big-Five personality and security awareness

Earlier psychological research established a link between cyber security behavior and the Big-Five personality, the characteristics of which are summarized in Table 1 (*Gosling, Rentfrow & Swann, 2003*). For instance, *Russell et al. (2017)* found negative correlations between emotional instability (neuroticism) and secure cyber behaviors (*e.g.*, using protection software against malware and virus) and between conscientiousness and insecure cyber behaviors (*e.g.*, using unsecured wireless networks), whereas *Shappie, Dawson & Debb (2020)* revealed that in addition to conscientiousness; agreeableness and openness positively predict cybersecurity behaviors (*e.g.*, keeping anti-virus software up to date). Meanwhile, *Kennison & Chan-Tin (2020)* reported rather different results: extraversion, agreeableness, and emotional stability -not openness nor conscientiousness- explain why some users are prone to commit risky cyber behaviors (*e.g.*, not signing out of a shared computer, sharing password with someone else) while others are not.

It appears that the relationships between Big-Five personality traits and cybersecurity awareness seem to vary across contexts and depend on the indicators measured in the study. However, in terms of SSO security awareness, and also by taking into consideration the Indonesian context as a collectivist country, we argue that being extraverted and agreeable is associated with lower SSO security awareness. Users having these traits are more likely to share their passwords with someone else, either voluntarily out of trust or when asked by others they respect or fear due to social status. On the other hand, we argue that being conscientious, emotionally stable, and open is associated with higher SSO security awareness. Users having these traits are arguably more cautious in their decision making and thus will avoid risky behavior with their SSO accounts. Thus, our hypotheses are as follows:

**Table 1  Big-Five personality.**

| Personality | Traits |
|---|---|
| Extraversion | Enthusiastic, not reserved, extraverted, not quiet |
| Agreeableness | Not critical, sympathetic, warm, not quarrelsome |
| Conscientiousness | Organized, careful, dependable, self-disciplined |
| Emotional stability | Not anxious, not easily upset, calm, emotionally stable |
| Openness | Creative, not conventional, open to new experience, complex |

H6: Extraversion negatively predicts SSO awareness

H7: Agreeableness negatively predicts SSO awareness

H8: Conscientiousness positively predicts SSO awareness

H9: Emotional stability positively predicts SSO awareness

H10: Openness positively predicts SSO awareness

Furthermore, since past studies reported significant correlations between agreeableness and privacy concerns (*Korzaan & Boswell, 2008*), and between conscientiousness and privacy concerns (*Junglas, Johnson & Spitzmüller, 2008*), we thus expect the aforementioned variables will interact with each other in predicting SSO security awareness. As such, our two final hypotheses are as follows:

H11: Agreeableness interacts with privacy concerns in predicting SSO security awareness

H12: Conscientiousness interacts with privacy concerns in predicting SSO security awareness

## MATERIALS & METHODS

### Participants

After obtaining approval from the Directorate of Research and Community Services within the university (No: 01.A/DirDPPM/70/DPPM/I/2021), we sent out a link to an online survey through broadcast email and WhatsApp messages to the academic community at one of the largest private universities in Indonesia, which has approximately 23,000 students and 1,000 faculty and staff. Between April 16 and May 4, 2021, a total of 283 participants ranging from 17 to 59 years of age (M = 26.63, SD = 10.23) completed the survey after providing their consents. The questionnaire was delivered in Bahasa Indonesia (see Supplementary Materials). To ensure eligibility and avoid duplication, all participants were required to use their SSO accounts to access the survey. More information about the demographics of respondents is available in Table 2.

### Measures

Apart from the three demographic variables (*i.e.,* gender, age, and academic role), there are three independent variables (*i.e.,* SSO familiarity, privacy concerns, and Big-Five personality) and one dependent variable (SSO account security awareness) in this study. Table 3 summarizes the variables of interest along with their respective measurement items that we developed for this study as follows:

**Table 2** Demographic information of all participants (n=283).

| Variable | Frequency | Percentage |
|---|---|---|
| **Gender** | | |
| Male | 148 | 52% |
| Female | 135 | 48% |
| **Age** | | |
| ≤19 years old | 72 | 25% |
| 20–29 years old | 132 | 47% |
| 30–39 years old | 38 | 13% |
| 40–49 years old | 31 | 11% |
| ≥50 years old | 10 | 4% |
| **Academic role** | | |
| Student | 197 | 70% |
| Faculty member | 34 | 12% |
| Staff | 52 | 18% |

*SSO Familiarity.* We developed three items to measure how well users are familiar with the SSO system in their university. The three items cover their overall knowledge of SSO along with its features and risk. We then aggregated the three items to calculate a composite score of SSO familiarity in the range of 0 to 100.

*Privacy Concerns.* We adopted privacy concerns scales developed by *Buchanan et al. (2007)* to measure user privacy concerns in this study. Specifically, we included only five items related to user accounts. We also calculated a composite score of privacy concerns in the range of 0 to 100 by aggregating all five items.

*Big-Five Personality.* We adopted the Ten Item Personality Inventory (TIPI), a very brief measure of the Big-Five personality domains (*Gosling, Rentfrow & Swann, 2003*), which has been widely used by many researchers in need of short personality measures in the past two decades. Specifically, we used the one that has been translated and validated in Bahasa Indonesia (*Hanif, 2018*) in this study.

*SSO Account Security Awareness.* We developed five items for each one of the Knowledge, Attitude, and Behavior dimension, resulting in a total of 15 items to measure SSO Account Security Awareness in this study by adapting the Human Aspects of the Information Security Questionnaire (HAIS-Q) (*Parsons et al., 2017*). We then calculated a composite score of SSO account security awareness in the range of 0 to 100 by using the weighted average method (30% for Knowledge, 20% for Attitude, and 50% for Behavior) to be classified further into three categories, *i.e.,* "Poor" (<60), "Average" (60–79.99), and "Good" (≥80) as recommended by *Kruger & Kearney (2006)*.

## Data analysis

We employed hierarchical linear regression in R 3.6.3 to analyze the data. As illustrated in Fig. 1, we conducted three steps of regression analysis with some additional independent variables in each model. In the first regression, we included only SSO familiarity and privacy concerns in addition to the three demographic variables (*i.e.,* gender, age, and academic role) as the predictors. Next, we introduced the Big-Five personality variables

**Table 3 Variables of interest and measurement items.**

| Variable | Code |
| --- | --- |
| **Familiarity with SSO** | **F** |
| 1. I know what the university's SSO account is. | F1 |
| 2. I know what systems and data are accessible with my university's SSO account. | F2 |
| 3. I am aware of the risk of negative impacts if my university's SSO account is used by other people. | F3 |
| **Privacy concerns** | **Pr** |
| 1. In general, how concerned are you about your privacy while you are using the internet? | Pr1 |
| 2. Are you concerned about online organizations not being who they claim they are? | Pr2 |
| 3. Are you concerned about online identity theft? | Pr3 |
| 4. Are you concerned about people online not being who they say they are? | Pr4 |
| 5. Are you concerned that an email you send may be read by someone else besides the person you sent it to? | Pr5 |
| **Knowledge** | **K** |
| 1. Using the same password for the university's SSO account and other personal accounts like social media is not prohibited. | K1r* |
| 2. Sharing my password for the university's SSO account to other people, including friends or colleagues, is not prohibited. | K2r* |
| 3. A combination of uppercase, lowercase, numbers, and special characters is a must when choosing password, including for the university's SSO account. | K3 |
| 4. Using a password that is 8 characters long or shorter is not prohibited. | K4r* |
| 5. When signing-in to the university account through the SSO system on a device that is not my own, using the incognito or private mode in the web browser is necessary. | K5 |
| **Attitude** | **A** |
| 1. It is safe enough to use the same password for the university's SSO account and other personal accounts like social media. | A1r* |
| 2. Sharing my password for the university's SSO account to other people, including friends or colleagues, is a bad idea. | A2* |
| 3. It is safe enough to use a password that consists of a combination of only alphabets, including for the university's SSO account. | A3r* |
| 4. It is safe enough to use a password that is 8 characters long or shorter, including for the university's SSO account. | A4r* |
| 5. Signing into the university's SSO account on a device that is not my own without using the incognito or private mode in the web browser is risky. | A5 |

**Table 3** (*continued*)

| Variable | Code |
|---|---|
| **Behavior** | **B** |
| 1. I use a different password for the university's SSO account than my other personal accounts like social media. | B1 |
| 2. I share my password for the university's SSO account with friends or colleagues at the university. | B2r* |
| 3. I use a combination of uppercase, lowercase, numbers, and special characters for all my passwords, including the university's SSO account. | B3 |
| 4. I always use passwords that are more than 8 characters long, including for the university's SSO account. | B4 |
| 5. I hardly ever use incognito or private mode in the web browser when signing into the university's SSO account on a device that is not my own. | B5r* |
| **Security Awareness Score** | **Score** |

**Notes.**
    * Reverse items were inverted prior to calculation.

to the model in the second regression. Finally, we added the interaction terms between privacy concerns and two out of five Big-Five personality variables (*i.e.,* agreeableness and conscientiousness) in the third regression following the link between them as shown in the literature (*Osatuyi, 2015*). Additionally, we ran several diagnostic tests on the regression model (*i.e.,* Residual Plot, Normal Q-Q Plot, Scale-Location Plot, and Cook's distance) to look for potential outliers where we identified three influential cases that we then omitted prior to repeating the hierarchical regression analysis. The dataset and the corresponding R code for analysis are available on our GitHub repository (see Data Availability section).

# RESULTS

The summary statistics are provided in Table 4 for the dependent variable and in Table 5 for the independent variables. As can be seen, the average SSO security awareness score for all participants in this study is 69.31 out of 100, which falls into the "Average" category according to the rubric by *Kruger & Kearney (2006)*. When considering each individual measurement item, the mean for the majority of items is indeed between 60 and 79.99. Certain items relating to password reuse (K1, A1), password length (K4, A4), and the use of incognito mode on a shared device (B5) are classified as "Poor", while others relating to account sharing (K2, A2, B2) and password complexity (K3) are classified as "Good". Applying the same categorization to SSO familiarity (*i.e.,* 80.86 out of 100) and privacy concerns (*i.e.,* 85.90 out of 100), however, means they both fall into the "Good" category.

    The scatterplots in Fig. 2 illustrate how SSO security scores vary by demographic variables. As can be seen, the SSO security awareness scores and age tend to form a negative linear relationship. This relationship is typically consistent across genders and academic roles.

    Additionally, the dumbbell plots in Fig. 3 indicate that SSO security awareness is relatively consistent across genders, but not across academic roles. Students consistently demonstrated significantly lower levels of knowledge, attitude, and behavior regarding

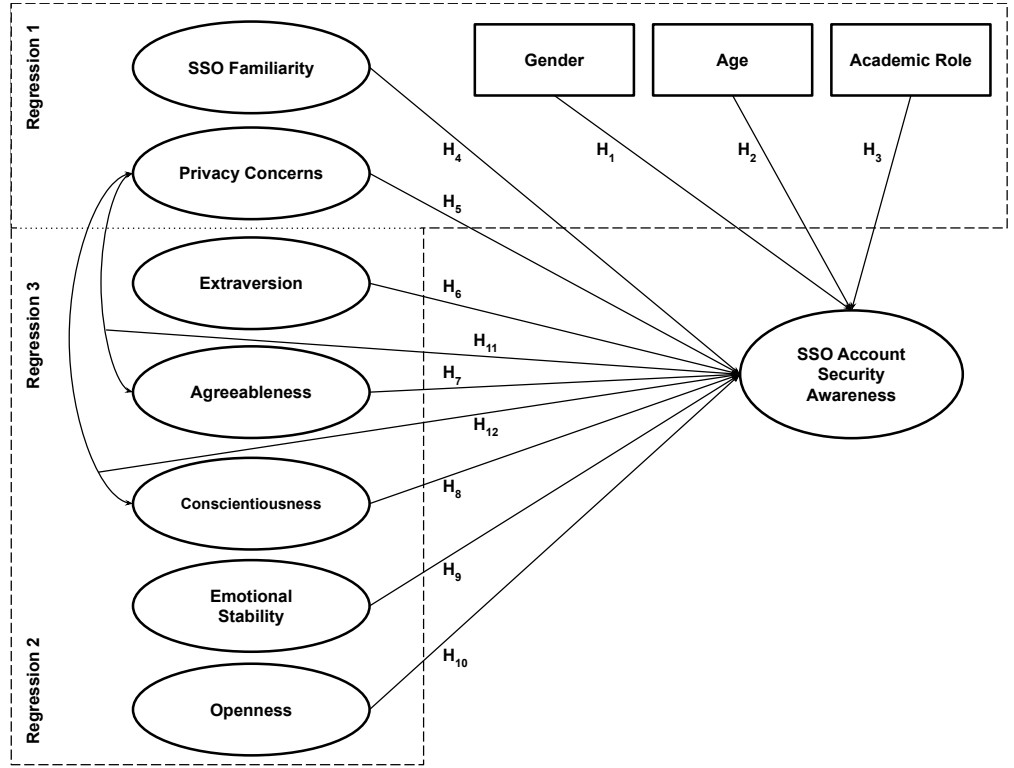

**Figure 1  Conceptual model of SSO account security awareness in this study.**

SSO account security compared to faculty and staff. Apart from their score in attitude that is much lower and closer to student's score, staff scored fairly close to faculty in terms of knowledge and behavior, resulting in no significant differences in total score between the two.

Following that, Table 6 summarizes the results of the hierarchical regression analysis. As can be seen, all independent variables, with the exception of gender, were found to be statistically significant in the first regression and they remained significant in the second regression after the addition of Big-Five personality traits as independent variables in the model. While only one of the five personality traits (*i.e.,* extraversion) was found to be statistically significant in the second regression, the addition of interaction terms between some traits (*i.e.,* agreeableness and conscientiousness) and privacy concerns in the third regression altered the result. As it turned out, statistical significance was found for all but one of the Big-Five personality traits (*i.e.,* openness). With the addition of these interaction terms, another significant finding emerged: privacy concerns were no longer significant predictors of SSO security awareness on their own. Rather than that, they interact with agreeableness and conscientiousness, as depicted in Fig. 4.

**Table 4  Summary statistics of the dependent variable.**

| Variable | Mean | SD | Min | Max |
|---|---|---|---|---|
| **Knowledge (0–100)** | **66.91** | **16.26** | **25** | **100** |
| K1: password reuse | 47.00 | 34.59 | 0 | 100 |
| K2: sharing SSO account | 82.86 | 26.00 | 0 | 100 |
| K3: password complexity | 84.28 | 25.18 | 0 | 100 |
| K4: password length | 46.38 | 33.32 | 0 | 100 |
| K5: incognito mode | 74.03 | 26.07 | 0 | 100 |
| **Attitude (0–100)** | **62.69** | **18.90** | **15** | **100** |
| A1: password reuse | 51.50 | 33.32 | 0 | 100 |
| A2: sharing SSO account | 80.83 | 28.97 | 0 | 100 |
| A3: password complexity | 60.51 | 30.11 | 0 | 100 |
| A4: password length | 42.84 | 31.29 | 0 | 100 |
| A5: incognito mode | 77.74 | 24.35 | 0 | 100 |
| **Behavior (0–100)** | **73.41** | **14.96** | **25** | **100** |
| B1: password reuse | 77.56 | 27.37 | 0 | 100 |
| B2: sharing SSO account | 86.31 | 23.45 | 0 | 100 |
| B3: password complexity | 78.45 | 24.72 | 0 | 100 |
| B4: password length | 75.00 | 26.46 | 0 | 100 |
| B5: incognito mode | 49.73 | 31.54 | 0 | 100 |
| **Composite Score (0–100)** | **69.31** | **13.62** | **34.5** | **100** |

**Table 5  Summary statistics of the independent variables.**

| Variable | Mean | SD | Min | Max |
|---|---|---|---|---|
| **Familiarity with SSO (0–100)** | **80.86** | **18.56** | **25** | **100** |
| F1: know what SSO is | 82.60 | 20.19 | 0 | 100 |
| F2: know what systems and data are accessible with SSO | 77.12 | 23.06 | 0 | 100 |
| F3: aware of the risk of SSO account being used by others | 82.86 | 23.40 | 0 | 100 |
| **Privacy Concerns (0–100)** | **85.90** | **14.54** | **30** | **100** |
| P1: general privacy concerns on the Internet | 79.95 | 20.46 | 0 | 100 |
| P2: false identity of organizations online | 84.72 | 20.59 | 0 | 100 |
| P3: online identity theft | 84.28 | 20.20 | 0 | 100 |
| P4: false identity of other individuals online | 93.11 | 14.78 | 0 | 100 |
| P5: confidentiality of messages | 87.46 | 19.57 | 0 | 100 |
| **Big-Five personality (1–7)** | | | | |
| Extraversion | 4.14 | 1.18 | 1.00 | 7.00 |
| Agreeableness | 5.30 | 1.03 | 1.00 | 7.00 |
| Conscientiousness | 5.14 | 1.05 | 2.50 | 7.00 |
| Emotional Stability | 4.71 | 1.20 | 2.00 | 7.00 |
| Openness | 5.33 | 1.07 | 1.50 | 7.00 |

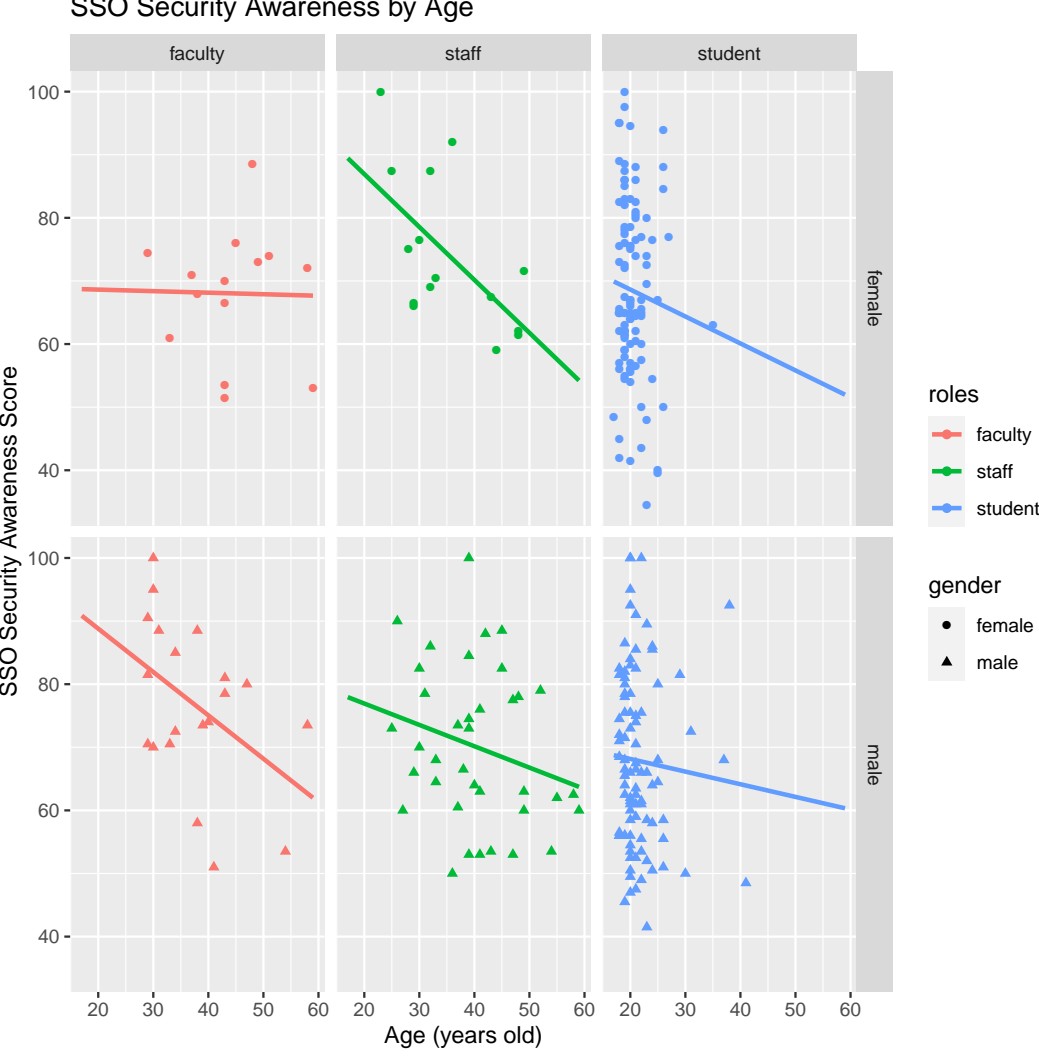

**Figure 2  Scatterplots of SSO security awareness score by age, gender, and academic roles.**

Furthermore, as shown in Table 6, these interaction terms are the two strongest predictors in the final model based on their standardized coefficients.

Taking all of the preceding findings into account, Table 7 summarizes the results of hypothesis tests and Fig. 5 illustrates the final model based on those findings.

# DISCUSSION

## Demographics and SSO security awareness

As illustrated in Fig. 2 and futher confirmed in Table 6, the generational gap remains present in this study. Older users have lower security awareness than their younger colleagues. While arguments from prior research that older people did not get the same chance as younger people did in terms of digital literacy exposure including cyber security measures (*Grimes et al., 2010*) may still hold water, this finding brings a more serious issue in this

**SSO Account Security Awareness by Gender and Academic Role**

**Figure 3** Dumbbell plots of SSO security awareness by gender and academic roles.

study's context. In Indonesia, which may also be true in many other countries, age arguably correlates with job seniority position. Putting this into the context of SSO, therefore, the older the users, the more systems and information are at risk should any cybersecurity incidents happen. Linking with our previous argument that users of the same educational setting should arguably receive similar exposure, it could be the case that such programs used by the IT department in introducing SSO technology and its security may not well address their older users yet. In other words, they seem to work effectively only for younger generations. The fact that SSO familiarity significantly predicts SSO awareness further supports our argument.

Interestingly, our analysis results confirmed all demographic hypotheses except for gender. As illustrated in Fig. 3, the security awareness scores are nearly identical between males and females. Even more so, the association between gender and SSO security awareness remains non-significant after putting psychological factors as well as their interaction terms with privacy concerns into the equation as shown in Table 6. These unexpected findings contradict past research reporting that female users tend to be less aware of cyber security measures (*Anwar et al., 2017*; *Pratama & Firmansyah, 2021*). Considering that this study takes place within a single higher education institution, it could be the case that both male and female users have already been exposed to similar levels of SSO usage within their institution. Ergo, such gender distinctions have no bearing on their security awareness. The absence of statistically significant differences in SSO security awareness between males and females in this study is encouraging because it demonstrates that organizations can rely on both male and female users having the same level of SSO

**Table 6  Hierarchical regression analysis.**

| Predictor variables | Regression 1 | | | | Regression 2 | | | | Regression 3 | | | |
|---|---|---|---|---|---|---|---|---|---|---|---|---|
| | B | SE B | β | *p* | B | SE B | β | *p* | B | SE B | β | *p* |
| Constant | 63.37 | 8.68 | – | <.001 | 67.83 | 9.49 | – | <.001 | 71.46 | 25.67 | – | .006 |
| Gender (Male) | −0.14 | 1.55 | −0.01 | .931 | −0.77 | 1.57 | −0.03 | .625 | −0.82 | 1.54 | −0.03 | .592 |
| Age | **−0.36** | **0.14** | **−0.28** | **.010** | **−0.34** | **0.14** | **−0.26** | **.013** | **−0.35** | **0.14** | **−0.27** | **.011** |
| Academic Role * | | | | | | | | | | | | |
| -Staff | −2.40 | 2.80 | −0.07 | .391 | −2.77 | 2.77 | −0.08 | .317 | −2.87 | 2.73 | −0.08 | .294 |
| -Student | **−13.79** | **3.50** | **−0.48** | **<.001** | **−12.88** | **3.46** | **−0.45** | **<.001** | **−13.25** | **3.42** | **−0.46** | **<.001** |
| Familiarity with SSO Account | **0.11** | **0.04** | **0.16** | **.007** | **0.10** | **0.04** | **0.14** | **.023** | **0.10** | **0.04** | **0.14** | **.024** |
| Privacy concerns | **0.15** | **0.06** | **0.17** | **.007** | **0.15** | **0.06** | **0.16** | **.010** | 0.11 | 0.29 | −0.12 | .692 |
| Big-Five Personality | | | | | | | | | | | | |
| -Extraversion | | | | | **−1.47** | **0.66** | **−0.13** | **.027** | **−1.31** | **0.65** | **−0.12** | **.046** |
| -Agreeableness | | | | | −0.93 | 0.86 | −0.07 | .282 | **−16.11** | **5.04** | **−1.22** | **.002** |
| -Conscientiousness | | | | | 1.57 | 0.87 | 0.12 | .074 | **15.43** | **5.41** | **1.21** | **.005** |
| -Emotional Stability | | | | | 1.37 | 0.79 | 0.12 | .085 | **1.63** | **0.79** | **0.15** | **.038** |
| -Openness | | | | | −0.73 | 0.83 | −0.06 | .385 | −0.73 | 0.82 | −0.06 | .378 |
| Interaction Terms | | | | | | | | | | | | |
| -Privacy concerns x Agreeableness | | | | | | | | | **0.17** | **0.06** | **1.56** | **.003** |
| -Privacy concerns x Conscientiousness | | | | | | | | | **−0.16** | **0.06** | **−1.42** | **.011** |
| Observations | 280 | | | | 280 | | | | 280 | | | |
| df | 273 | | | | 268 | | | | 266 | | | |
| *p* | <.001 | | | | <.001 | | | | <.001 | | | |
| R² | 0.125 | | | | 0.170 | | | | 0.201 | | | |
| Adjusted R² | 0.106 | | | | 0.136 | | | | 0.162 | | | |
| δR² | - | | | | 0.045 | | | | 0.031 | | | |
| δAdjusted R² | - | | | | 0.030 | | | | 0.026 | | | |

**Notes.**

    * Faculty member is used as the reference category; numbers reported are unstandardized coefficients (B), standard errors of unstandardized coefficients (SE B), standardized coefficients (β), and p-values (p); the bold and blue numbers denote statistically significant values ($p < .05$).

security awareness. It could also be attributed to the organization's success in educating users regardless of their gender.

    On the other hand, past research indicating gender gap in cybersecurity awareness either took place in a workplace in which participants' chances to get exposed to such cyber measures might vary (*Anwar et al., 2017*) or their participants came from different places altogether (*Pratama & Firmansyah, 2021*). Those having IT related backgrounds and working directly with external clients might be more aware of cyber threats compared to those having no IT backgrounds and working with internal clients only. Considering that females are still underrepresented in IT related jobs (*Richter, 2021*), it could be the main reason why such a gender gap existed in past research. As such, we argue that any study revealing a gender disparity in cybersecurity awareness should delve deeper into the reason for it than simply gender.

    As also illustrated in Fig. 3 and confirmed in Table 6, we found that students are less aware of SSO security than staff and faculty, who share similar SSO security awareness level. On one hand, it can be the case because they have less things to lose if such incidents happen. As *Pratama & Firmansyah (2021)* argue, how sensitive people are to cyber threats and how

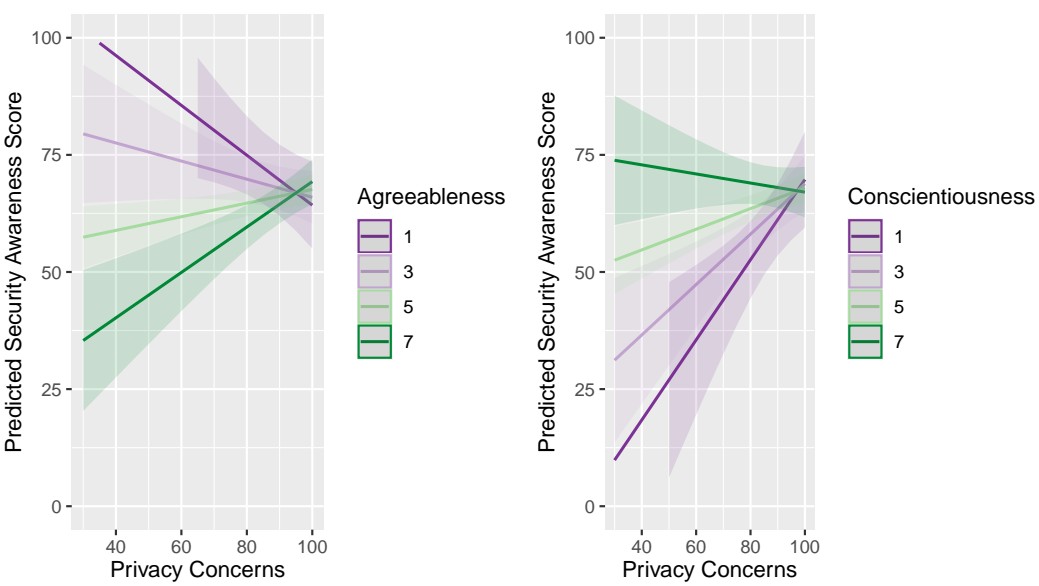

**Figure 4** **Marginal effects of the interaction terms between Big-Five personality traits and privacy concerns.**

**Table 7 Summary of hypothesis tests results.**

| Hypothesis | Relationship | Result |
|---|---|---|
| H1 | Females are less aware of SSO security | Not supported |
| H2 | Older people are less aware of SSO security | Supported |
| H3 | Students are less aware of SSO security than faculty/staff | Supported |
| H4 | Familiarity to SSO positively predicts SSO security awareness | Supported |
| H5 | Privacy concerns positively predict SSO security awareness | Partially supported |
| H6 | | Supported |
| H7 | Extraversion negatively predicts SSO awareness | Supported |
| H8 | Agreeableness negatively predicts SSO awareness | Supported |
| H9 | Conscientiousness positively predicts SSO awareness | Supported |
| H10 | Emotional stability positively predicts SSO awareness | Not supported |
| H11 | Openness positively predicts SSO awareness | Supported |
| | Agreeableness interacts with privacy concerns in predicting | |
| H12 | SSO security awareness | Supported |
| | Conscientiousness interacts with privacy concerns in predicting SSO security awareness | |

well they adhere to cyber security measures is directly proportional to the magnitude of their potential losses should such incidents occur. Students, arguably, have less to lose in regard to their SSO account. On the contrary, faculty and academic staff have a plethora of sensitive data at risk, ranging from financial and salary information to any other private or confidential data, both to them as users and to their institution. As such, it is unsurprising that faculty and staff are more cognizant of SSO account security than students are. On the

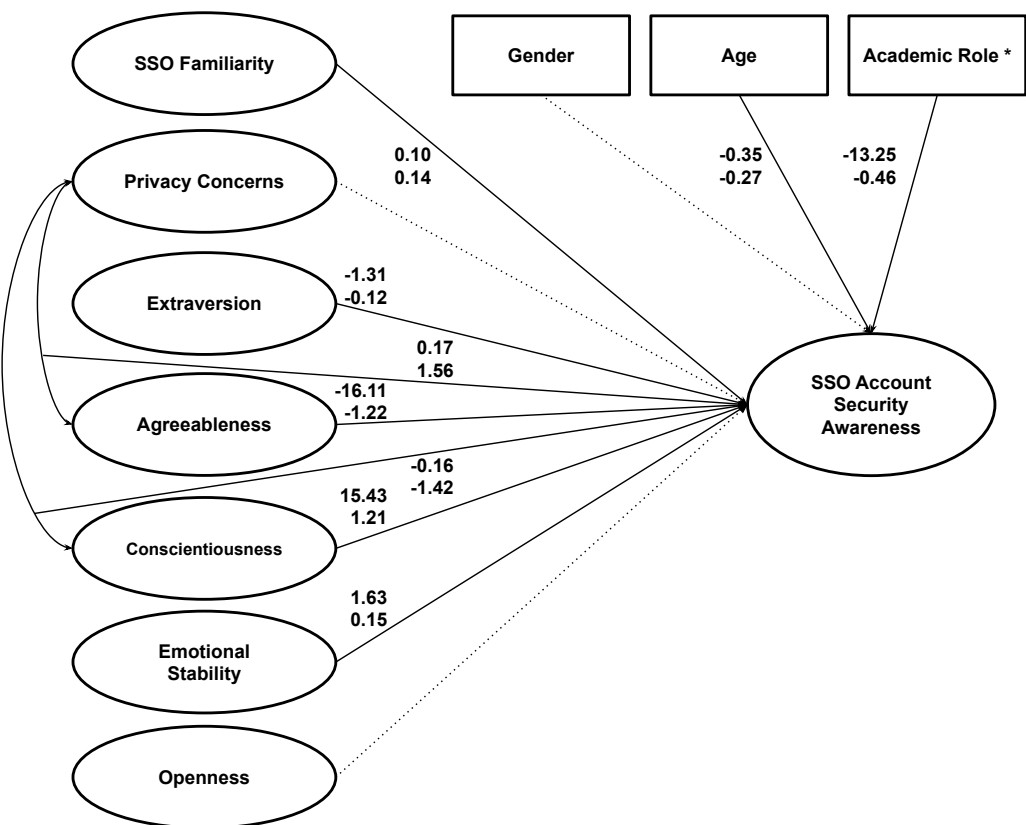

**Figure 5** **The final model of SSO account security awareness in this study.** * students compared to faculty members; Solid line indicates statistically significant relationship at *p* < .05; Dotted line indicates not statistically significant relationship was found; Numbers reported are the unstandardized coefficient (top) and the standardized coefficient (bottom).

other hand, the fact that students are significantly less aware of SSO account security also leads to another suspicious behavior. Some students might intentionally share their SSO accounts. While it would be unethical to suspect all students, given that academic cheating is perceived as collaboration in collectivistic cultures such as Indonesia (*Jamaluddin, Adi & Lufityanto, 2021*), the possibility that a few students misuse their SSO accounts by intentionally sharing them with each other for academic dishonesty cannot be ruled out.

## Privacy concerns, Big-Five personality, and SSO security awareness

As expected, privacy concerns are positively associated with SSO security awareness, at least in the first two regressions as shown in Table 6. Holding all other variables constant, the more concerned users are about their privacy, the more aware they are of SSO security. Interestingly, when we factor in their interaction with the Big-Five personality constructs, this association becomes irrelevant. In other words, the extent to which privacy concerns may affect users' SSO security awareness is determined by their personality traits. Users with a high degree of agreeableness (*i.e.,* warm, not critical) are generally less aware of SSO security, but this would change if they also had increased privacy concerns. On the

contrary, users who are naturally conscientious (*i.e.,* organized, cautious) tend to have a high level of security awareness regarding their SSO accounts regardless of their privacy concerns, even if the latter can help those with a low degree of conscientiousness improve their security awareness. In this regard, regardless of their level of privacy concerns, it is the users' personality that naturally compels them to be more circumspect and critical, thereby increasing their awareness of the risks associated with their SSO accounts.

In contrast to the two aforementioned traits, extraversion and emotional stability account for SSO security awareness in ways that go beyond privacy concerns. In this regard, the more extraverted users are, the less aware they are of SSO security. By contrast, users who are emotionally stable are more likely to be aware of SSO security. Interestingly, even after controlling for demographic and privacy concerns variables, only the openness trait has no significant association with SSO security awareness. Our attempt to determine whether there are any interactions between these three characteristics and privacy concerns, which revealed none, confirms that these findings are robust.

These findings altogether suggest the needs of a tailored approach should interventions be designed to increase users' SSO security awareness. For example, intervention emphasizing privacy risk may work best for users with a high degree of agreeableness but is less efficient for users with high degree of conscientiousness. While for users with a high degree of emotional stability, it may be better to teach them about SSO security measures. A particular attention should be paid to users with extraverted traits given the negative association with security awareness in the model. Some conventional interventions may not work as effectively as it is for other personality traits. Perhaps, such further behavioral interventions may be needed. We suggest future research to explore this area more to shed light on different types of education and interventions that can work better for different types of personality traits.

## CONCLUSIONS

Our study discovered unique relationships between SSO security awareness, demographic characteristics, privacy concerns, and personality traits. The degree to which users are aware of SSO security varies according to their demographic characteristics and is determined in part by their personality traits, some of which (*i.e.,* agreeableness and conscientiousness) interact with their level of privacy concerns. The absence of gender disparities in SSO security awareness in this study suggests that it is possible to close the gap under the right circumstances, including but not limited to education and policy. It also suggests that closing generational gaps may be a greater challenge than closing gender gaps in cybersecurity awareness.

## FUTURE WORK

This study lays the groundwork for future research and interventions aimed at increasing user awareness of SSO security and closing any existing gaps between different demographic groups of users, particularly in higher education settings. An experimental study examining various types of intervention on SSO security awareness is one way to accomplish this.

With the growing adoption of SSO by colleges and universities worldwide, addressing this issue of SSO security awareness is becoming increasingly important. Also, to gain a more holistic understanding of security awareness and practices surrounding SSO, we propose that researchers conduct similar studies in other parts of the world, taking into account cultural differences that may affect cybersecurity awareness, particularly regarding SSO security.

## ACKNOWLEDGEMENTS

We would like to express our gratitude to Dr. Mukhammad Andri Setiawan and his team at the Universitas Islam Indonesia's Board of Information Systems Board of Information Systems for providing technical information about the university's SSO implementation, and to the Universitas Islam Indonesia's Division of Public Relations Division of Public Relations for assisting us with data collection.

### Funding

This work was supported by Universitas Islam Indonesia under the Directorate of Research and Community Services research grant (No. 010/Dir/DPPM/70/Pen.Unggulan/XII/2020), and the Directorate of Academic Development's publication grant. The funders had no role in study design, data collection and analysis, decision to publish, or preparation of the manuscript.

### Grant Disclosures

The following grant information was disclosed by the authors:
Universitas Islam Indonesia under the Directorate of Research and Community Services research grant: No. 010/Dir/DPPM/70/Pen.Unggulan/XII/2020.
The Directorate of Academic Development's publication.

### Competing Interests

The authors declare there are no competing interests.

### Author Contributions

- Ahmad R. Pratama conceived and designed the experiments, performed the experiments, analyzed the data, performed the computation work, prepared figures and/or tables, authored or reviewed drafts of the paper, and approved the final draft.
- Firman M. Firmansyah conceived and designed the experiments, analyzed the data, authored or reviewed drafts of the paper, and approved the final draft.
- Fayruz Rahma conceived and designed the experiments, authored or reviewed drafts of the paper, and approved the final draft.

### Data Availability

The raw survey data consisting of demographics, academic roles, privacy concerns, SSO familiarity, Big-Five personality, and security awareness measures are available in the Supplemental File.

The data is also available at GitHub: https://github.com/ahmadrafie/ssostudy.

## Supplemental Information

Supplemental information for this article can be found online at http://dx.doi.org/10.7717/peerj-cs.918#supplemental-information.

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
