# Peer review of "Security awareness of single sign-on account in the academic community: the roles of demographics, privacy concerns, and Big-Five personality"

_PeerJ Computer Science, doi:10.7717/peerj-cs.918_

## Round 0.1 · original submission · Minor Revisions

Add a paper structure paragraph at the end of the Introduction Section. Add the future research directions section. Ensure your bibliography is up-to-date.

Reviewer 1 ·

Basic reporting

The paper is well written and is easy to follow. The results are reported with professional diagrams and tables.

Improvement suggestion:
1. The figures should be vector graphics. I suggest the authors regenerate them into pdf or ps graphics
2. Citations from top security conferences (e.g., NDSS, IEEE S&P) are largely missing. The authors are suggested to look through literatures starting with the following paper: AuthScan: Automatic Extraction of Web Authentication Protocols from Implementations

Experimental design

The paper manages to make clear the hypotheses, and design use studies to validate them. I am very happy to see the experiments well follow the ethics.

Improvement suggestion:
1. The privacy concerns (Table 3) in the questionnaire seem not complete. For example, the relying party (RP) may access user's data on identify provider (IDP). It would be good if the authors may consider to include these.

Validity of the findings

The conclusions are well stated, and backed by the user studies.

·

Basic reporting

Thank you so much for the editor for giving me an opportunity to review this article.

Very well written article in Professional English. The research is clear and unambiguous, and very relevant to current topic of information security.

SSO is an area where a lot of security attacks can happen in organizations. Educational sector often overlooks security and privacy of information. This article is timely to elaborate with appropriate data about the security mindset of participants in the educational institution.

Please to verify spacing and formatting issues before sending the final copy.

Experimental design

The design and reserach fits within the Information Security domain

The research is well defined and attached sumary and survey results fills in the idenfied knowledge gap of SSO.

Validity of the findings

Data has been well analysed and conclusions are linked to original research questions.

Reviewer 3 ·

Basic reporting

There are many papers in the usability security literature that focus on 2FA evaluations, for example Das “Why Johnny Doesn’t Use Two Factor A Two-Phase Usability Study of the FIDO U2F Security key - https://link.springer.com/chapter/10.1007%2F978-3-662-58387-6_9 and since the paper mentions 2FA a couple more well known references could be included. There is a similar paper by the same principal author related to older users and this could provide a complement to the Pratama and Firmansyah paper mention on lines 85-86.

The authors mention privacy concerns and it would be good to make reference to https://cups.cs.cmu.edu/soups/2007/posters/p173_heckle.pdf Heckle and Lutters, SOUPS 2007.

The level of the grammar and flow of the paper was good.
The strength of the paper is the way that they have drawn broadly from the literature since there is not a great deal of literature on SSO evaluations in higher education

Experimental design

It would have been interesting to see a hypothesis which took into account whether the subjects they were studying had an impact on their awareness, for example if they were from an arts background or a science background. This related to line 142 so it would be interesting why they chose that hypothesis and now one which was linked to their academic backgrounds?
So for example in Figure 3 the security awareness was compared to behaviour attitude and knowledge. I would like to have seen more explanation as to why this aspect was not expanded on more to include the type of faculty they were related to. If some of the staff were professional services staff then it would have been good to have split the categorisation of people more finely. It was not entirely clear what the difference was between students, faculty and staff were on line 73.
The methodology did not indicate on line 188 what the size of the overall population was and what were the limitations of the survey. Did everyone complete all the answers and so were the answers normalised?

Validity of the findings

Thank you for providing the tables at the back of the articles but in the main body of the discussion no specific numbers were presented which related to the tables. I would have expected the discussion section to make reference to the tables.

The commentary in the discussion did match the tables but the narrative would have been strengthened with more specific referencing to the tables.

·

Basic reporting

Background information is covered pretty well. Please see attached PDF for specific comments though.

Experimental design

Rationale and procedures seems good. Please see attached PDF for specific comments about how some of the DV's and IV's are being reported though.

Validity of the findings

Findings seem to be on target. Suggest modifying some of the figures and note that two figures are not mentioned in text. Also, some of the implications that pertain directly to personality (Big 5) are not well developed and could be made stronger. Please see attached PDF for specific comments though.

Additional comments

Please see attached PDF for specific comments. One of the biggest issues is to make sure the article gets proofread by a native English speaker to ensure grammatical consistency.

---

## Round 0.2 · accepted · Accept

Thank you for revising the article as per the reviewers' recommendations. The revised version is up to the journal standard.

Reviewer 1 ·

Basic reporting

The paper is well written and is easy to follow. The results are reported with professional diagrams
and tables.

Experimental design

The paper manages to make clear the hypotheses, and design use studies to validate them.

Validity of the findings

The conclusions are well stated, and backed by the user studies.

Additional comments

Thank the authors for the revision. I am OK for this version to be published.

·

Basic reporting

The authors fixed the changes mentioned by the reviewers.

Experimental design

The authors fixed the changes mentioned by the reviewers.

Validity of the findings

The authors fixed the changes mentioned by the reviewers.

Additional comments

None